# Copper and Zinc Removal Efficiency of Two Reactive Filter Media Treating Motorway Runoff—Model for Service Life Estimation

Raúl Rodríguez-Gómez [1], Agnieszka Renman [1,*], Batoul Mahmoudzadeh [2] and Gunno Renman [1]

1   Division of Water and Environmental Engineering, KTH Royal Institute of Technology, SE-100 44 Stockholm, Sweden; raulr@kth.se (R.R.-G.); gunno@kth.se (G.R.)
2   Sverige Consulting Group, Ramboll Water, SE-104 62 Stockholm, Sweden; batoul.mahmoudzadeh@ramboll.se
*   Correspondence: agak@kth.se; Tel.: +46-8-7906768

**Abstract:** The predominant techniques used for road runoff treatment are sedimentation and filtration. In filtration systems, the ability of the media to adsorb the contaminants is a finite process. Consequently, construction, operation and maintenance managers of such systems should know in advance the service life, i.e., when the used medium should be replaced, and associated costs of operation and maintenance. A batch experiment followed by a packed bed reactor (PBR) experiment addressed the kinetics of the studied media argon oxygen decarburization slag (AOD) and Polonite, followed by the development of a 1D-model to describe the change of concentration of Cu and Zn within time. The batch test results showed that Cu and Zn adsorption followed the Freundlich isotherms for AOD and Polonite. Those results coupled with the linear driving force model and the developed model resulted in good agreement between the PBR results and the simulation. The model was capable to predict (i), the service life at the hydraulic load of 0.18 m/h for AOD (Cu: 395 d; Zn: 479 d) and Polonite (Cu: 445 d; Zn: 910 d), to show (ii) the profile concentration in the PBR within time and the gradient of the concentration along the height of the reactor.

**Keywords:** 1D-model; AOD; packed bed reactor; Polonite; stormwater; simulation

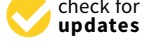

## 1. Introduction

The pollutants transported by runoff from motorways and roads with intense vehicular traffic has been a subject of study in many countries over the past four decades (e.g., [1–3]). The effects caused by the road runoff pollution in the aquatic environment (e.g., [4,5]) and solutions to overcome environmental problems have been studied [6]. Copper (Cu) and zinc (Zn) commonly occur in the highest concentrations in runoff from motorways in Sweden [7–10]. These two metals mostly occur in the dissolved phase and are considered relevant in runoff from traffic areas during all seasons because of their total concentrations, fractionation behavior, and toxicity [11]. The European Union Water Framework Directive states that all bodies of water must achieve good conditions [12]. This means that discharged stormwater should not affect the quantity, quality, or ecology of waterbodies. According to the Federal Soil Protection Law in Germany the maximum allowable pollutant concentrations for Cu and Zn discharged after a treatment device should not exceed 50 and 500 µg/L respectively [13]. In Sweden, national limit values have not yet been set for emissions of metals via stormwater. However, so-called regional background values calculated for lakes and watercourses in the country's seven eco-regions can be used as guideline values for the substances released after stormwater treatment. A regional background value is typical for water that is unaffected by point-source pollution or acidification, however, may have been affected by the deposit of combustion residues from fossil and other fuels as well as diffuse emissions from agriculture and forestry. Such values, for e.g., metals, are considered rather as reference values for the current situation in the recipient water [14].

In order to meet the regulatory water quality criterium set by environmental authorities, it is important to design adequate purification systems and know their service life [11]. Sedimentation and filtration are the predominant treatment techniques used where open ponds, wetlands and soil infiltration frequently represents these steps [15,16]. Dissolved metals usually are to be removed by chemical precipitation or retained by reactive filter media [17]. The use of road salting during winter in cold climate regions can however desorb and mobilize metals bound in filter media [18]. In cities, such as Stockholm where space is often limited, the Swedish Road Administration has implemented underground facilities with sedimentation and the addition of flocculent as a treatment method. Where space is available, for example at traffic junctions, open sedimentation ponds are built. However, stormwater needs further treatment after sedimentation in order to meet more stringent limit values of e.g., heavy metals. The use of filter materials is thereby one option, in the case that they remove pollutants efficiently and are suitable for operation and cost-efficient maintenance [19–22]. The latter aspects are critical to the client, less so than the actual construction and cost of the facility. For the producer of any filter material, it should be mandatory to test its removal capacity and longevity by an independent party, i.e., through certification or a similar procedure. Although the importance of filter longevity in constructed wetlands and other passive treatment systems was highlighted 20 years ago [23], the service life is however rarely considered in research papers investigating filter materials for water treatment.

A technique to solve this issue is to perform laboratory batch and column experiments or pilot-scale trials to evaluate the performance of filter materials. However, such experiments can be time-consuming and expensive especially if a column trial is run until the break-through of studied elements [23]. Usually, synthetic stormwater is used, which will not reflect in situ conditions with real road runoff. In order to shorten the length of the experiment, a modeling tool can help to predict the life of a certain filter media. However, any model input data must be as realistic as possible in terms of hydraulic loading rates, filter media particle size and composition of the stormwater used in the experiment [24–26]. Jourak et al. [27] executed the convection-dispersion equation in the HYDRUS-1D software to predict the phosphate breakthrough curve for columns filled with the reactive filter material Filtra P. They concluded that the model was suitable for systems fed with a low influent concentration of phosphate, which probably would be valid for different metals. McKenzie et al. [28] developed a model in MATLAB where the soil-water distribution coefficient and the ratio of the drainage area to the infiltration-based stormwater treatment area, i.e., loading ratio (LR) was used for the simulation. The hydrogeologic and solute transport components were then decisive for the model output. Huber et al. [11] suggested a standardized method to determine the service life of filter materials used in decentralized stormwater treatment systems. Their method consists of several steps and considers preloading the filter media in a pilot-scale model with copper and zinc by a load of n-1 years of the estimated service life (n). They concluded that column experiments should be preferred to batch experiments to determine efficiencies and service lifespans.

Rodríguez-Gómez and Renman [29] coupled a transport equation with the Langmuir isotherm to predict the behavior of a packed bed reactor (PBR) filled with the reactive filter material Polonite. Their model predicted the percentage removal of concentration of phosphorus within time and its profile along the height of the reactor. In the present study, we applied the PBR technique combined with batch experiments to generate data for a dynamic model, aiming to describe processes responsible for the life span of the selected filter media argon oxygen decarburization slag (AOD) and Polonite. Steel slag and other alkaline media have been found to have a high potential for several environmental applications, such as filters in stormwater treatment, due to their chemical and physical properties [30–32].

The use of a PBR constructed as a column may be a good approach to describe the contaminant removal mechanisms and hydraulics associated with stormwater treatment. This is valid for systems that use a filling medium i.e., any porous filter media. We

conducted the experiment with real stormwater from a motorway and measured Cu and Zn, metals that constantly occur in high concentrations in road runoff. The aim was to develop a dynamic model to describe the processes occurring in the PBR. Our ultimate goal was to develop a tool for estimation of the service life of filter media with AOD-slag and Polonite as the first tested filter materials.

## 2. Materials and Methods

### 2.1. Characterization of Filter Materials and Stormwater Used in the Experiments

AOD slag and Polonite were obtained from the company Outokumpu Stainless AB and Polonite Nordic, respectively. They were used as a test medium for the batch and PBR experiments. AOD Slag and Polonite® are commercially available in the standard size fraction of 2–6 mm. However, prior to conducting the experiments, they were sieved to obtain a particle size distribution $d_{50}$ of 3.1 mm and with the range 2.0–4.0 mm to fit the size of the PBR (see Section 2.3). Density and porosity were determined to 1180 kg/m$^3$ and 0.595, respectively for AOD and 730 kg/m$^3$ and 0.45 respectively for Polonite. The AOD has the following mineralogical composition: CaO, 49%; Al$_2$O$_3$, 21%; SiO$_2$,19%; MgO, 8%; FeO, 0.7%; Cr$_2$O$_3$, 0.8%; MnO, 0.4%; CaF$_2$, 0.8% and TiO$_2$, 0.3% and Polonite has: CaO, 40%; Al$_2$O, 4%; SiO$_2$, 40%; MgO, 0.7%; Fe$_2$O, 2%; K$_2$O, 0.7% and Na$_2$O 90%. Fresh AOD and Polonite are alkaline materials having a pH of 12.6 and 12.5, respectively (measured in distilled water, *w/v* 1:2.5). Real runoff used for the experiments was collected in 30 L plastic containers every second week from the Fredhäll stormwater treatment plant located in central Stockholm [33] and transported to the laboratory where it was stored in cold conditions (4–6 °C). The treatment plant receives stormwater from the E4 motorway with a current Annual Average Daily Traffic (AADT) at this section of 140,000 vehicles. The speed limit is 70 km/h. The stormwater used in the laboratory experiments has undergone chemical flocculation and sedimentation in the plant. However, the treated water still has high dissolved concentrations of Cu and Zn, which corresponds to that usually found in stormwater [2]. The concentrations of Cu and Zn were however in the batch experiments increased from about 20 and 60 μg/L to 100 and 200 μg/L respectively to increase the accuracy and validity of the batch results, since stormwater not only contains Cu and Zn. The metal salts used for spiking the stormwater were CuSO$_4$ 5 H$_2$O (Purity: 99%; Brand: MERCK) and ZnCl$_2$ (Purity: 98%; Brand: KEBO Lab).

### 2.2. Batch Experiment

Batch experiments were performed to determine the isotherms governing the adsorption of Cu and Zn onto AOD. The filter material dosage was 0, 0.5, 1.0, 1.5 and 3 g and each mass was added to 50 mL of the spiked runoff to 100 mL Erlenmeyer flasks for each of the four contact times 24, 42, 48 and 55 h. A single solution was prepared for each element and the experiment was performed at room temperature (20–22 °C). A horizontal platform shaker with a speed of 150 rpm was used to agitate the solution with its respective media (i.e., AOD or Polonite). The Cu and Zn equilibrium concentrations were determined after the four contact times with an inductively coupled plasma (Thermo ICP iCap 6000) equipment. Finally, the Langmuir and Freundlich isotherms were tested and established based on the results from the batch experiment and used as input in the developed 1D-model.

### 2.3. Packed Bed Reactor Experiment

The PBR consisted of a transparent polyvinyl chloride column with a height of 22 cm and a diameter of 3.3 cm. Five columns were constructed and filled with different filter materials of which two were AOD slag and Polonite. The results of the three other material's performance will be published elsewhere. The ratio of the inner diameter to mean particle diameter ($d_{50}$) was slightly more than 10:1 in which the wall effect can be negligible [34]. The raw stormwater was pumped continuously from the influent storage tank (IST) using an ISMATEC IP tubing peristaltic pump. A slow-motion stirrer operated 30 min every hour

kept the stormwater with an average turbidity of 100 NTU in mixed conditions. The flow rate was 0.331 L/d, equal to a hydraulic load of 0.16 m/h (according to a recommendation by the Swedish Transport Administration), and the operation time was 18 weeks. The hydraulic retention time was approximately 5.6 h. The effluent placed at the top of the column had a hose that diverted the treated stormwater to the effluent storage tank (EST). Samples were taken weekly from the tanks and the concentration of Cu and Zn was determined using the ICP-MS instrument. The PBR was under operation from late April to early September 2020, in total 126 days. A schematization of the PBR experiment is shown in Figure 1.

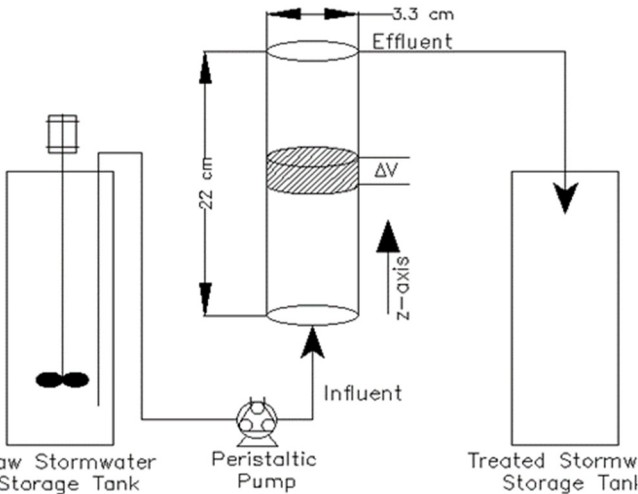

**Figure 1.** Schematization of the PBR. $\Delta V$ = Control volume.

*2.4. Conceptual Model*

The PBR is operated in up-flow mode and the expected removal of Cu and Zn follows a gradient from high in the inlet zone towards low in the outlet. Consequently, the concentration of Cu and Zn decreases along the height of the PBR. The expected mechanisms taking place in the z-axis of the PBR are diffusion, advection and adsorption. The effluent from where the stormwater comes out with a lower concentration of Cu and Zn compared to the influent is located at the top of the PBR.

A mass balance for the concentration of Cu or Zn in the PBR (Figure 1) is written as:

$$\frac{\partial C_{ij}}{\partial t} = D_{ij}\frac{\partial^2 C_{ij}}{\partial z^2} - v_{ij}\frac{\partial C_{ij}}{\partial z} - \frac{\rho_{ij}}{\varepsilon_{ij}}\frac{\partial q_{ij}}{\partial t} \tag{1}$$

where $C$ is the concentration of the element under study (i.e., Cu or Zn) in mg/L. $t$ is the time in units of days. $D$ is the diffusion coefficient in m²/s. $z$ represents the axial coordinate (i.e., the height of the PBR). $v$ is the up-flow velocity in m/s. The medium (i.e., AOD or Polonite) bulk density is given in kg/m³ and porosity is represented by $\rho$ and $\varepsilon$, correspondingly. $q$ denotes the concentration of the element under study in the solid phase at equilibrium (mg/g). The subscript $i$ may be AOD or Polonite and $j$ represents Cu or Zn depending on the parameter that is analyzed. In Equation (1) the left-hand side term describes the change of concentration of Cu or Zn within time. On the right-hand side, the first and second terms are the transports terms by diffusion and advection, respectively. The third term represents the kinetic occurring in the media, which involves an adsorption process. The change of concentration of Cu or Zn in the solid phase (i.e., $\partial q_i/\partial t$) may be expressed by the linear driving force model [35], which can be written as:

$$\frac{\partial q_{ij}}{\partial t} = k_{ij}\left(q_{fij} - q_{ij}\right) \tag{2}$$

where $k_{ij}$ symbolizes the adsorption rate constant of pseudo-first-order in (1/s) and $q_{fij}$ is the adsorption capacity at time $t$ in units of mg/g. The parameter $q_{ij}$ is described by the Freundlich adsorption model (Equation (3)), which is the isotherm that fits the batch experiment better (see Section 3.1 Batch Experiment). However, the Langmuir model could describe the adsorption capacity in cases that the batch experiment followed Langmuir isotherm.

$$q_{ij} = K_{f_{ij}} * C_{eij}^{1/n} \qquad (3)$$

where $K_{f_{ij}}$ is the Freundlich capacity factor that relates the amount of heavy metal absorbed by the media with the amount of heavy metal contained in the stormwater, and n is Freundlich intensity parameter that is an indicator of the heterogeneity of the media.

The initial condition for Equation (1) may be written as:

$$C_{(z,t=0)} = 0 \qquad (4)$$

For Equation (2) the initial condition is:

$$q_{ij(z,t=0)} = 0 \qquad (5)$$

Which means that at time zero, the adsorption process of Cu or Zn has not started yet (i.e., new media).

The boundary conditions for Equation (1) are:

$$-D_{ij} \left. \frac{\partial C_{ij}}{\partial z} \right|_{(z=0)} = v_{ij}\left(C_{0ij} - C_{ij}\right) \qquad (6)$$

$$\left. \frac{\partial C_{ij}}{\partial z} \right|_{(z=L)} = 0 \qquad (7)$$

Equation (6) means that at the influent, the variation of the concentration of Cu or Zn along the height of the PBR is a function of the dispersion coefficient and flow velocity. While at the effluent of the PBR (Equation (7)), there is no variation of concentration of Cu or Zn.

The boundary conditions for Equation (2) are:

$$\left. \frac{\partial q_{ij}}{\partial z} \right|_{(z=0)} = 0 \qquad (8)$$

$$\left. \frac{\partial q_{ij}}{\partial z} \right|_{(z=L)} = 0 \qquad (9)$$

Equations (8) and (9) mean that both in the influent and in the effluent, there is no adsorption process since at those points there are no media.

*2.5. Simulation and Statistical Analysis*

The governing equations of the model (i.e., Equations (1)–(3)) may be solved using the function 1-D parabolic and elliptic partial differential equation in MATLAB Software. The parameters used to execute the simulation are present in Table 1. The diffusion coefficient was calculated using the definition of Peclet number [29]. Statistical analysis was performed using Microsoft data analysis toolpak excel 365.

**Table 1.** Parameters used for the simulation.

| Parameter | Symbol | Units | AOD | Polonite |
|---|---|---|---|---|
| Height of the PBR | $z$ | m | 0.22 | 0.22 |
| Diameter of the PBR | $\varnothing$ | m | $3.3 \times 10^{-2}$ | $3.3 \times 10^{-2}$ |
| Stormwater velocity | $v$ | m/s | $1.35 \times 10^{-3}$ | $1.35 \times 10^{-3}$ |
| Porosity | $\varepsilon$ | | 0.595 | 0.45 |
| Density | $\rho$ | $Kg/m^3$ | 1180 | 730 |
| Operation time | $t$ | d | 126 | 126 |
| Peclet number | $Pe$ | | 10 | 10 |
| Diffusion coefficient | $D$ | $m^2/s$ | $D = v*z/Pe$ | $D = v*z/Pe$ |

## 3. Results and Discussion

### 3.1. Batch Experiment

The Langmuir and Freundlich isotherms were found to be able to describe the adsorption kinetics of Cu and Zn in AOD and Polonite. Table 2 shows the kinetic parameters of Freundlich and Langmuir isotherm obtained from the batch experiment for Cu and Zn. Clearly, the Freundlich isotherm dominates the adsorption mechanisms in the batch experiment. This is known by comparing the coefficient of determination (i.e., $R^2$) for the Freundlich and Langmuir isotherm.

**Table 2.** Adsorption isotherm parameters obtained from the batch experiment.

| Filter Material [Metal] | Freundlich Parameter | | | Langmuir Parameter | | |
|---|---|---|---|---|---|---|
| | $K_f$ [L/g] | n | $R^2$ | a [mg/g] | b [L/mg] | $R^2$ |
| Polonite [Cu] | 0.1813 | 1.0801 | 0.9665 | 0.0549 | 4.8564 | 0.131 |
| AOD [Cu] | 0.0099 | 3.3069 | 0.8591 | 0.0043 | 40.485 | 0.4305 |
| Polonite [Zn] | 0.0613 | 1.5827 | 0.9794 | 0.0211 | 16.8472 | 0.8115 |
| AOD [Zn] | 0.1018 | 1.2552 | 0.9751 | 0.0393 | 6.4368 | 0.6924 |

Figure 2 shows the correlation between the batch experiments and the trend line and its respective equation from which the Freundlich constant was derived. Neither Freundlich nor Langmuir showed a negative slope. This means that there was no repulsive interaction between adsorbate and absorbate during the batch experiment [36,37]. The slope (1/n) in the dominant isotherm (i.e., Freundlich) ranged from 0 to 1. This indicates that the adsorption of the heavy metal in the media is due to chemical bonds (i.e., chemisorption), implying a heterogeneous surface of the studied media [38].

Since in the Freundlich isotherm model, the amount of heavy metal adsorbed may be infinite as the concentration of heavy metal increases in the liquid phase, we can say that a limit in adsorption capacity cannot be defined (cf. [39]). However, some researchers have reported $K_f$ as maximum sorption capacity (e.g., [40,41]) and others just as adsorption capacity [42]. However, as stated previously in this paper, the Freundlich isotherm (Equation (3)) is used here as a kinetic term in the developed model when Cu and Zn interact with AOD and Polonite.

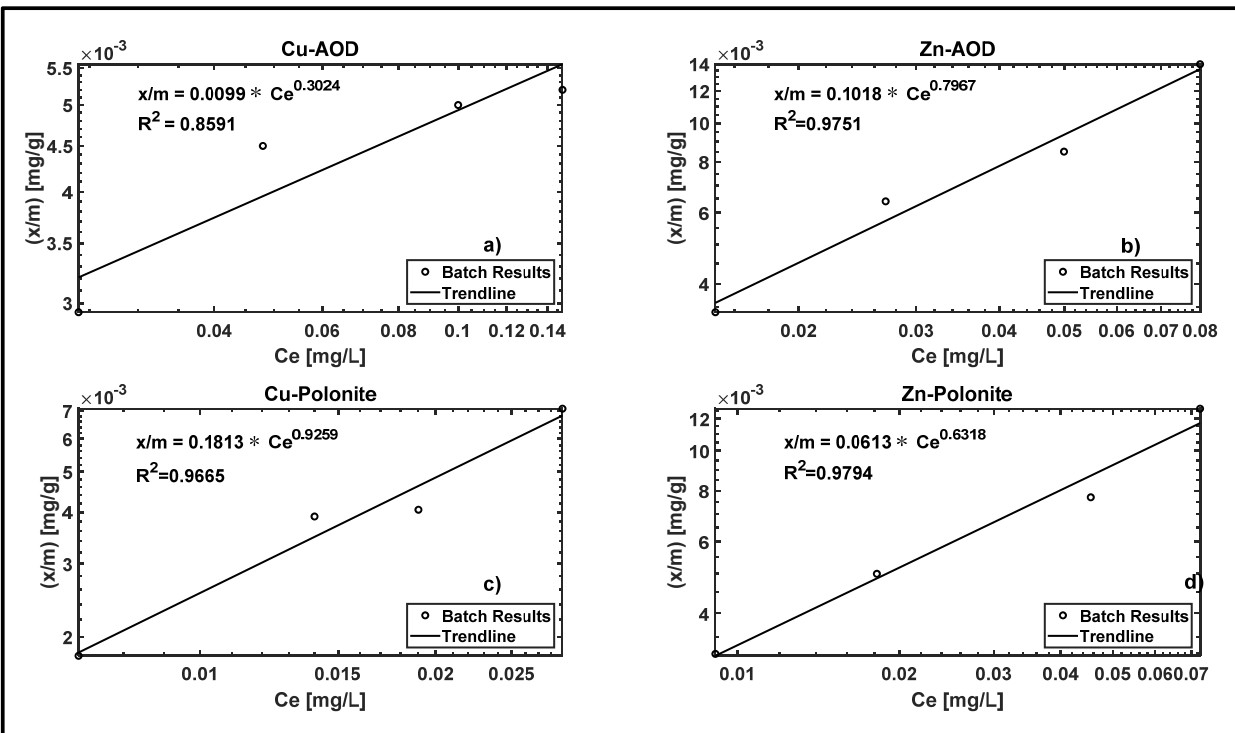

**Figure 2.** Correlation between batch results and trend line. $C_e$: Equilibrium concentration (mg/L), x/m: Adsorption ratio (mg adsorbate/g media). (**a**) Freundlich isotherm for Cu and AOD, (**b**) Freundlich isotherm for Zn and AOD, (**c**) Freundlich isotherm for Cu and Polonite, and (**d**) Freundlich isotherm for Zn and Polonite.

### 3.2. Packed Bed Reactor

A road runoff volume of 42 L was pumped to each PBR, corresponding to 223-bed volumes. The pH of effluent decreased from 12.7 at the start, to 11.6 at the end of the trial. Figure 3 shows the dissolved influent and effluent concentrations of Cu and Zn for AOD and Polonite during the 18 weeks of operation time.

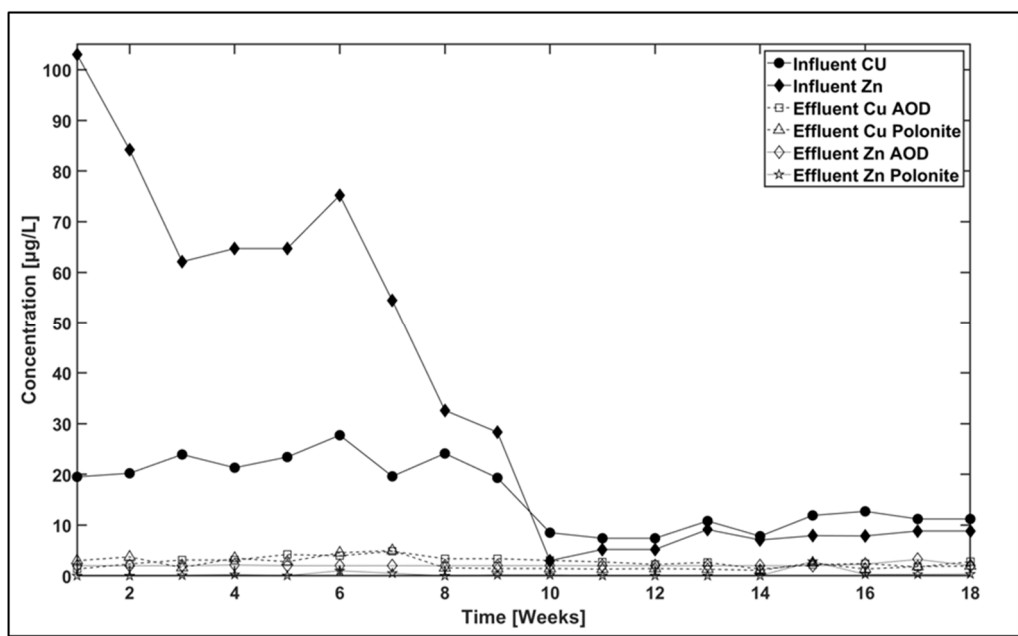

**Figure 3.** Temporal changes in dissolved influent and effluent Cu and Zn concentrations in the PBR experiment.

The regularly collected stormwater varied in heavy metal concentration and showed the highest values during the first 10 weeks of the experimental period. Spring and early summer periods may have runoff with elevated heavy metal concentrations that have not yet been washed away from traffic areas, where accumulation has occurred during the winter period [33]. The concentration of Cu in the influent was on average 16 µg/L and the effluent concentration of Cu for AOD and Polonite were 2.80 µg/L and 2.30 µg/L, respectively. For Zn, the average concentration of the influent was 30 µg/L and the average effluent concentration of Zn for AOD and Polonite were 2.09 µg/L and 0.34 µg/L, respectively. The material's removal efficiency did not vary much over time, although the effluent concentration was much higher at the beginning than at the end of the experiment (Figure 3). Effluent pH decreased from 12.61 to 11.56 and 12.00 to 9.36 for AOD and Polonite, respectively. The unchanged removal efficiency can be explained by the high pH of the percolating stormwater above the solubility point, which causes metals to precipitate, most likely as metal oxides and metal carbonates [43]. Table 3 shows a comparison between the percentage removal of Cu and Zn for AOD and Polonite.

**Table 3.** Average removal efficiencies of the studied filter media in the PBR experiment.

| Filter Material [Metal] | Influent [µg/L] | Effluent [µg/L] | Removal [%] |
|---|---|---|---|
| AOD [Cu] | 16 | 2.80 | 82 |
| Polonite [Cu] | 16 | 2.30 | 86 |
| AOD [Zn] | 30 | 2.09 | 92 |
| Polonite [Zn] | 30 | 0.34 | 99 |

A statistical analysis ($\alpha = 0.05$) comparing AOD and Polonite performed to the eighteen data for Cu and Zn showed that for Cu there is no significant difference in the effluent concentration of AOD and Polonite. For Zn, however, the statistical analysis showed a significant difference in the effluent concentration for AOD and Polonite. Several types of slag have been tested to remove heavy metals from stormwater. For instance, Trenouth and Gharabaghi [44] reported that the Cu concentration percentage removal from synthetic stormwater by blast furnace slag (BFS) and oxygenated furnace slag (OFS) are 90% and 95%, correspondingly. For Zn they reported a removal percentage of 84% and 61% for BFS and OFS, respectively. The AOD utilized in the present study showed better performance for removal of Zn but lower performance for Cu compared to those mentioned slags. The fact that real stormwater was used in our study, probably influenced the removal efficiency of the filter medium, since real stormwater contains compounds, such as suspended solids and microorganisms that could clog the PBR. Polonite has been widely studied for the treatment of domestic wastewater and particularly for phosphorus removal (e.g., [29,45–47]). For the treatment of heavy metals, however, the publications are limited [43,48]. Renman et al. [43] reported the percentage removal of Cu and Zn from domestic wastewater to 16% and 62%, correspondingly. Household wastewater has higher concentrations of organic solids and generally much lower metal content than stormwater, which can explain the difference in the sorption capacity of Polonite.

### 3.3. Simulation

Table 4 shows the parameters used in the model. The operation time was 126 days (i.e., 18 weeks), and the values utilized as the influent concentration for Cu and Zn were the mean values during the PBR experiment. The overall adsorption rate values for the column experiment were determined following the Froment and Bischoff procedure [49]. Complementary parameters are shown in Table 2 (Freundlich parameters) and Table 1 (Geometry and operation conditions of the PBR).

**Table 4.** Parameters used to run the model.

| Material | Cu Influent [μg/L] | Zn Influent [μg/L] | Cu Adsorption Rate [1/d] | Zn Adsorption Rate [1/d] |
|---|---|---|---|---|
| AOD | 16 | 30 | 0.18 | 0.35 |
| Polonite | 16 | 30 | 0.22 | 0.80 |

Figure 4 shows the model response for the concentration of Cu and Zn in the effluent of the PBR (a–d) and its corresponding percentage removal (a′–d′) for AOD and Polonite.

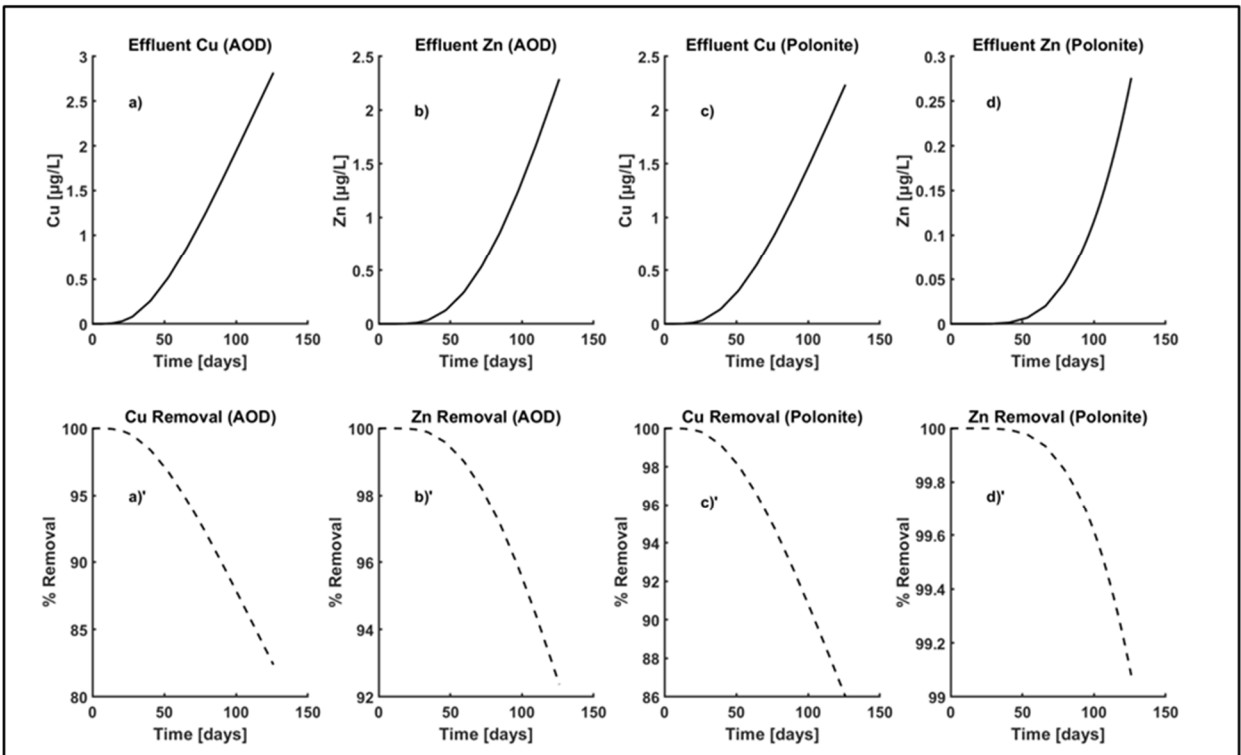

**Figure 4.** Transient state and percentage removal of the concentration of Cu and Zn in the PBR according to the modelling response. (**a**)′, (**b**)′, (**c**)′, and (**d**)′ correspond to the percentage removal of (**a**), (**b**), (**c**), and (**d**), respectively.

The simulation was in agreement with the experimental results. In both cases, Polonite showed better performance for the removal of Cu and Zn compared with AOD. Analyzing the concentration of Cu and Zn when Polonite is used as media, it is observed that Polonite has more affinity in the uptake of Zn than Cu. The developed model demonstrated that it can be used to predict the efficiency of the removal of the concentration of Cu and Zn in the PBR. However, it can provide additional information that could be complicated to obtain in the experiment. In the next section, the capability of the model is shown.

### 3.4. Capability of the Model

Besides the transient state within time, the model is able to display the life span of the filter media, the accumulation profile of the amount of the heavy metal in the solid phase (i.e., AOD or Polonite), and the concentration along the height of the PBR. As a demonstration, the Zn concentration will be used to show the capability of the model. All parameters remain the same, except for the operating time that is extended to such a value to obtain a zero-removal percentage, i.e., complete break-through.

### 3.4.1. Profile of Zn Concentration and Percentage Removal

Figure 5 shows the profile concentration and the percentage removal of Zn within time for AOD and Polonite. Despite that in Sweden there are no national limit values for the discharge of heavy metals from stormwater treatment plants, some municipalities have implemented limit values to protect waterbodies. For stormwater effluent, the Järfälla municipality in the Stockholm metropolitan area proposed a maximum limit value for the concentration of Zn of 15 µg/L and 9 µg/L for the concentration of Cu. The set limit value means that the filter mass must be changed when the outgoing water exceeds this value, i.e., the service life. The simulation showed a service life of 479 days for the AOD and 910 days for the Polonite according to Järfälla municipality guideline. The service life of the media may be prejudiced by other components presents in the stormwater, which can control and influence the result of the modeling. Joo et al. [50], Conley et al. [51] and Kandra et al. [52] reported clogging of filter media due to the concentration of suspended solids in the stormwater. The clogging phenomena have been also simulated in infiltration systems treating rainfall-runoff [53]. Wakida et al. [54] investigated the effects of solids in the treatment of stormwater of an urban area of 13 ha and they concluded the importance of pretreatment before subsequent filtration for the removal of dissolved metals. In the case of Cu, the simulation predicts a service life of 395 days for AOD and 445 days for Polonite to reach the Järfälla municipality guideline (i.e., 9 µg/L).

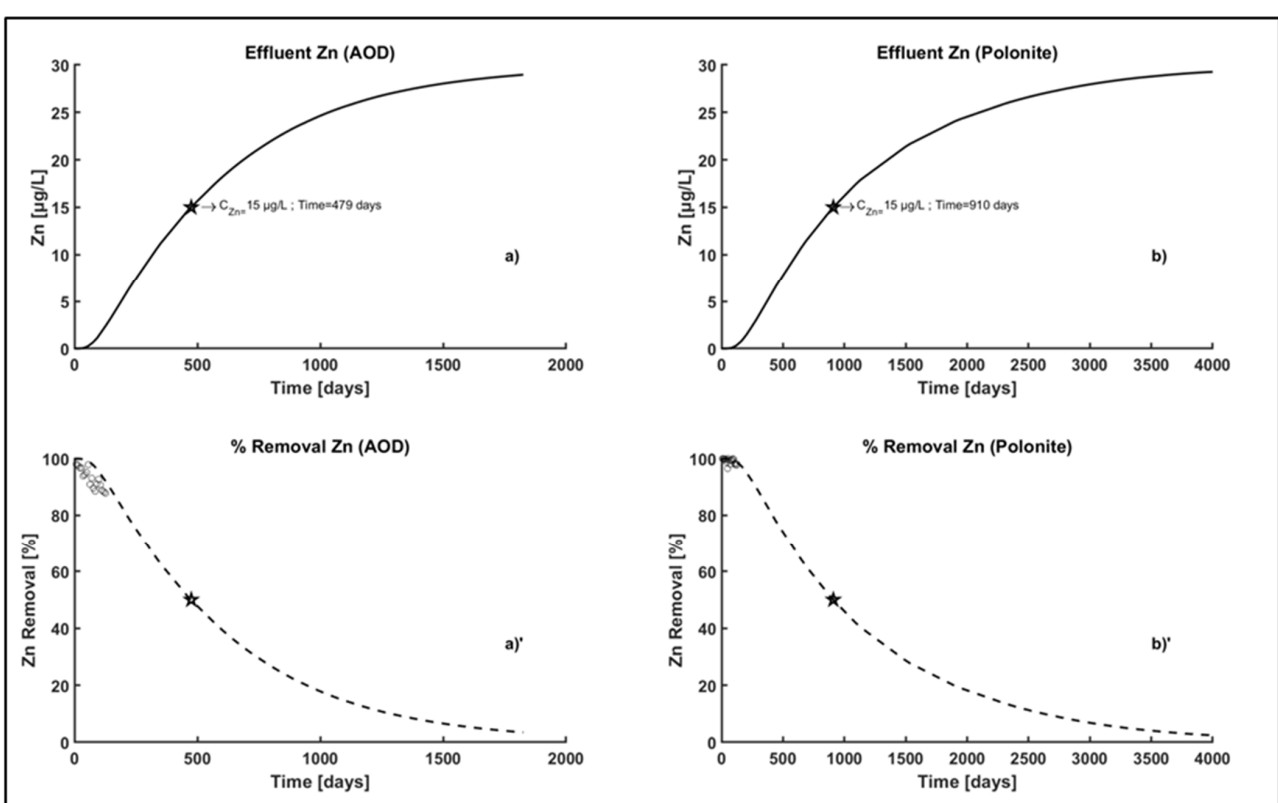

**Figure 5.** Capability of the model. Profile of the concentration and percentage removal of Zn within time for AOD and Polonite. Star indicates the maximum limit value according Järfälla municipality and circles refers to the experimental data. (**a**)′ and (**b**)′ are the percentage removal of Zn correspondingly to (**a**) and (**b**).

### 3.4.2. Profile of Zn Concentration

In this simulation, the operation time was set to 500 days to show the gradient of the concentration of Zn at different times. Figure 6a, b show the profile of the concentration of Zn in the PBR for AOD and Polonite, respectively. After 500 days, the concentration of Zn in the PBR (i.e., in the solid phase) is lower for the AOD than Polonite.

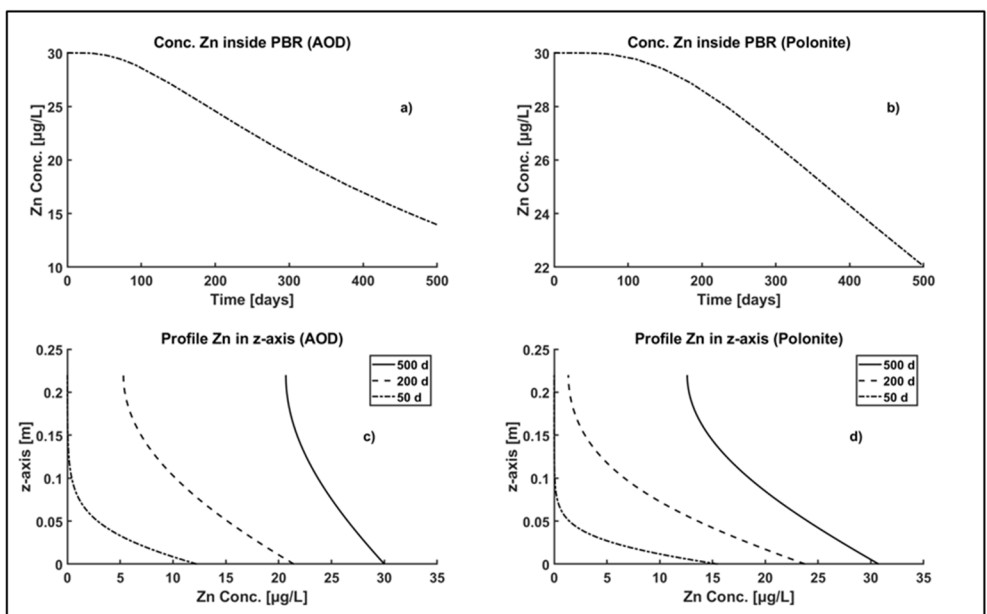

**Figure 6.** Comparison of the profile and gradient concentration of Zn for AOD and Polonite.

Figure 6c, d show the gradient of the concentration of Zn along the height of the PBR at different intervals of time for AOD and Polonite. Since the influent concentration of Zn is 30 μg/L with a constant flow, that value is the maximum concentration that can remain inside the PBR. Figure 6 shows how the profile of the concentration of Zn trend to 30 μg/L as time passes. The AOD media is reaching the maximum concentration value faster than Polonite. This is in accordance with Figure 6a since in both cases it is shown that the lower Zn adsorption capacity of AOD compared to Polonite results in a longer life span for Polonite than for AOD. A sensitive analysis performed by keeping all parameters constants except the parameter of interest showed that the model is sensitive to the flow velocity, which is directly related to the hydraulic retention time, adsorption kinetic parameters (i.e., Freundlich isotherm constants), and the overall adsorption rate.

## 4. Conclusions

The adsorption kinetics followed the Freundlich isotherm model for all cases and from which it was deduced that the adsorption process in AOD and Polonite is due to the heterogeneous surface of the media. Since the Freundlich isotherm model does not show the maximum adsorption capacity of the media, the overall adsorption rate was determined based on the PBR experimental results. Overall, the Polonite medium had superior performance compared to AOD for both Cu and Zn removal. The PBR experiment showed a high capacity of adsorption of Cu and Zn for AOD and Polonite. After 18 weeks, the concentration of Cu and Zn were still under the guideline values of a Stockholm metropolitan municipality for both media. The developed model that was used with input results from batch experiment and parameters from PBR experiment fitted very well with the experimental results; however, the PBR experimental data represented a much shorter time than the modeled time frame. The estimated service life is valid only for the stormwater hydraulic loading of 0.16 m/h; however, the model should be able to predict the life span at different loading rates.

**Author Contributions:** Conceptualization, G.R., R.R.-G. and A.R.; methodology, G.R., R.R.-G. and A.R.; software, R.R.-G. and A.R.; validation, R.R.-G., A.R. and G.R.; formal analysis, R.R.-G. and A.R.; investigation, R.R.-G., A.R., B.M.; resources, A.R.; data curation, R.R.-G.; writing—original draft preparation, R.R.-G. and A.R.; writing—review and editing, R.R.-G. and A.R.; visualization, R.R.-G. and A.R.; supervision, A.R. and G.R.; project administration, A.R.; funding acquisition, A.R. All authors have read and agreed to the published version of the manuscript.

**Funding:** This research was funded by KTH Royal Institute of Technology by Post Doctor fellowship to Raúl Rodíguez Gómez and was a part of research granted to Agnieszka Renman within projects financed by Jernkontoret and the Swedish Transport Administration.

**Institutional Review Board Statement:** Not applicable.

**Informed Consent Statement:** Not applicable.

**Data Availability Statement:** Not applicable.

**Acknowledgments:** The authors would like to thank the companies Outokumpu Stainless AB and Polonite Nordic for providing AOD slag and Polonite, respectively. Thanks are due to Magnus Hallberg, Swedish Transport Administration, for rewarding discussions and comments on the manuscript.

**Conflicts of Interest:** The authors declare no conflict of interest. The funders had no role in the design of the study; in the collection, analyses, or interpretation of data; in the writing of the manuscript, or in the decision to publish the results.

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
