# Peer review of "Copper and Zinc Removal Efficiency of Two Reactive Filter Media Treating Motorway Runoff—Model for Service Life Estimation"

_water, doi:10.3390/w13182592_

Round 1

Reviewer 1 Report

Manuscript Number: water-1362579

Title: Copper and zinc removal efficiency of two reactive filter media
treating motorway runoff – model for service life estimation

Type: Research article

Recommendation: Major Revision

Comments to authors: The authors have presented an interesting study about a batch experiment followed by a packed bed reactor (PBR) addressing the kinetics of the studied media argon oxygen decarburization slag (AOD) and Polonite. The authors also developed 1D-model to describe the change of concentration of Cu and Zn within time. Authors have presented robust results with logical explanation. The work can be considered for publication after addressing the following comments.

  1. Fig 3: The influent Cu and Zn concentration varies a lot and it doesn't seem to affect the effluent concentration, why is it so?

  1. Does pH have any effect on the adsorption?

  1. There is a need for the characterization of adsorbents.

  1. In the batch experiment, samples were constantly taken out, does this change in volume and ultimately the pH and concentration?

Reviewer 2 Report

It is an interesting work for road runoff treatment via filtration systems, but the rational of the experiment design need to be improved, and the comparison between experiment results and model results should be discussed in the manuscript, some comments as following.

Major comments,

  1. Line 124, the stormwater used in the experiments has undergoing pre-treatment in the plant, the difference with real motorway runoff should be considered, especially the particle form pollutants.
  2. Line 127-128, the concentration of Cu and Zn used in the experiments was multiply, which effect for the adsorption model should be considered.
  3. Line 152, hydraulic load is an important parameters for the experiments results, and it was effect by many factors in real rainfall process, such as media particle size, rainfall intensify, and so on, why select 0.331 L/d?
  4. Line 256, the pH of effluent over than 11, if the precipitation happened of heavy metals during the experiment?

Minor comments,

  1. Line 114, kg/m3, the number 3 should be superscript?
  2. Table 1, ‘stormwater velocity’ mean to ‘stormwater infiltration rate in the PBR’? How to kept same value in AOD and Polonite?
  3. Line 262,264, ‘metal’ refer to ‘heavy metal’?
  4. Figure 4, in the beginning stage of the effluent model results exist great difference with experiment results (Figure 3)?
  5. Line 372-374, For Zn, the adsorption rates is repeated described.

Round 2

Reviewer 1 Report

Authors have addressed my comments and the manuscript can be published now.